# Sexually Transmitted Infections: Usefulness of Molecular Methods for Microorganism Detection in Stored Sexual Assault Samples

**DOI:** 10.3390/ijms26178124

**Published:** 2025-08-22

**Authors:** Laura Cainé, Ana Eira, Jennifer Fadoni, Magda Franco, Helena Correia Dias, António Amorim

**Affiliations:** 1National Institute of Legal Medicine and Forensic Sciences, I.P., Centre Branch, 3000-548 Coimbra, Portugalmagda.i.franco@inmlcf.mj.pt (M.F.); maria.h.dias@inmlcf.mj.pt (H.C.D.); antonio.j.amorim@inmlcf.mj.pt (A.A.); 2Faculty of Medicine, Porto University, 4200-319 Porto, Portugal; 3LAQV&REQUIMTE, Laboratory of Applied Chemistry, Department of Chemical Sciences, Faculty of Pharmacy, University of Porto, 4050-313 Porto, Portugal; jennifer.n.fadoni@inmlcf.mj.pt; 4National Institute of Legal Medicine and Forensic Sciences, I.P., North Branch, 4050-202 Porto, Portugal; 5Research Centre for Anthropology and Health, Department of Life Sciences, University of Coimbra, 3000-456 Coimbra, Portugal; 6Laboratory of Forensic Anthropology, Centre for Functional Ecology, Department of Life Sciences, University of Coimbra, 3000-456 Coimbra, Portugal; 7Faculty of Sciences, Lisbon University, 1749-016 Lisboa, Portugal

**Keywords:** sexual crimes, sexually transmitted infections, forensic microbiology

## Abstract

Sexual assault is a global public health and human rights concern, with serious physical, psychological and reproductive consequences for survivors. Among these, sexually transmitted infections are particularly relevant due to their frequently asymptomatic nature and potential for long-term complications. The detection of sexually transmitted infections in forensic settings is crucial for clinical management of victims and for evidentiary support in forensic sexual crimes investigations. This study aimed to evaluate the applicability of real-time polymerase chain reaction for detecting *Chlamydia trachomatis*, *Trichomonas vaginalis*, *Neisseria gonorrhoeae*, and *Treponema pallidum* in biological samples collected from victims of sexual assault and stored under routine forensic conditions, in some cases, for up to 18 years. A total of 231 swabs from 116 individuals collected between 2004 and 2017 were analysed using real-time PCR with pathogen-specific primers and fluorescent probes. The analysis revealed 13 positive samples of *T. vaginalis* (5.6%) and 11 of *C. trachomatis* (4.8%). No positive results were obtained for *N. gonorrhoeae* or *T. pallidum*. These findings demonstrate the usefulness of real-time polymerase chain reaction for detecting sexually transmitted infections in long-term preserved forensic samples. Moreover, the ability to identify pathogen DNA in archived samples highlights the potential role of molecular diagnostics in the retrospective investigation of sexual crimes, including cold cases. It underscores the value of molecular methods as a complementary tool in forensic proceedings and survivor care.

## 1. Introduction

The World Health Organization (WHO) defines sexual violence as “any sexual act, attempt to consummate a sexual act, or other act directed against a person’s sexuality by means of coercion, by another person, regardless of their relationship to the victim and in any context. It includes rape, which is defined as penetration, by physical or other coercion, of the vulva or anus with a penis, other body part, or object” [1].

Sexual violence has profound and multifactorial implications for survivors’ health, affecting physical, reproductive, and psychological domains. The severity and nature of these effects vary across age groups and may present as both acute and chronic conditions [2,3]. Among the most significant long-term health outcomes are sexually transmitted infections (STIs), which rank among the most prevalent infectious diseases worldwide [4].

STIs are primarily transmitted through sexual contact and constitute a major global public health challenge [5]. These infections may result from bacterial, viral, or parasitic pathogens and can affect individuals regardless of age, gender, or sexual orientation. If left untreated, STIs may lead to serious complications such as infertility, chronic pelvic or genital pain, and increased vulnerability to coinfections [6].

*Chlamydia trachomatis* (*C. trachomatis*) and *Trichomonas vaginalis* (*T. vaginalis*) are among the most prevalent sexually transmitted pathogens worldwide. Both infections may be symptomatic or asymptomatic and are associated with significant reproductive health complications, including urethritis, pelvic inflammatory disease, infertility, and increased susceptibility to coinfections [7,8,9,10]. The frequently silent nature of these infections complicates timely diagnosis and highlights the importance of sensitive detection methods, particularly in forensic contexts where early intervention may not be possible.

According to the WHO, the global burden of untreated STIS remains substantial, largely attributable to treatment failures and the frequently asymptomatic presentation of these infections. It is estimated that approximately one million new cases of STIS occur daily worldwide, culminating in nearly 374 million new infections annually [1]. The prevalence and incidence of STIS vary depending on the causative pathogen and are significantly influenced by socioeconomic determinants, with elevated rates commonly observed in regions characterized by under-resourced healthcare infrastructures [11].

Biological samples of forensic significance are frequently encountered in cases of physical assault and hold particular importance in the investigation of sexual offences. Such biological evidence is critically relevant to forensic investigations, as it may occasionally constitute the sole proof of the occurrence of sexual assault and enable the identification of the perpetrator [12]. It is imperative that these samples are collected within specialized facilities dedicated to victims of sexual violence or in clinical environments where individuals can receive comprehensive medical care, psychological support, and screening for sexually transmitted infections [13].

Forensic evidence—including clothing, urine, oral cavity swabs, hand and nail scrapings, as well as non-intimate skin samples—can facilitate the identification of perpetrators, the exclusion of suspects, and contribute to the successful prosecution of criminal cases [14]. In both victims and suspects of sexual assault, the majority of biological samples are conventionally collected using sterile cotton swabs [14].

To preserve the evidentiary value, sample collection must be performed with meticulous care to ensure the integrity and forensic reliability of the biological materials. Such materials should be stored under dry, temperature-controlled conditions to prevent microbial proliferation and degradation of deoxyribonucleic acid (DNA) [13]. Improper collection or handling procedures may compromise the suitability of samples for forensic analysis [12]. Given that often only trace amounts of DNA are available, stringent precautions are essential to avoid contamination, guarantee proper collection, and maintain optimal storage conditions [15], thereby preserving both the quality and quantity of the genetic material [14].

The detection of STIS is of critical importance and can be performed using various methodologies, including polymerase chain reaction (PCR) in real-time. This technique is a modification of conventional PCR that enables continuous monitoring of the amplification process in real time, while simultaneously facilitating the synthesis of multiple copies of a targeted DNA region. Real-time PCR has a wide range of applications, including pathogen detection and quantification, species identification, among others [16]. This study aims to evaluate the utility of real-time PCR for the detection of *C. trachomatis* and *T. vaginalis* in biological specimens obtained from sexual assault victims and stored for up to 18 years, for some samples. By demonstrating the reliability of this approach, we seek to emphasize the importance of molecular diagnostics in the forensic investigation of sexual crimes. The present work is therefore particularly relevant, as it involves the analysis of real forensic samples collected from sexual assault victims and stored under routine casework conditions. This research will contribute to the development of the national scientific literature and to the improvement of diagnostic practices in forensic and clinical contexts.

## 2. Results

Among the studied samples, 13 tested positive for *Trichomonas vaginalis* (5.6%) and 11 for *Chlamydia trachomatis* (4.8%). No positive results were observed for *Neisseria gonorrhoeae* or *Treponema pallidum*. These outcomes are illustrated in Figure 1. In the case of *C. trachomatis*, two positive samples were collected 18 years prior to analysis, one 10 years ago, five 9 years ago, and three 8 years ago (Figure 1). For *T. vaginalis*, two positive cases corresponded to samples collected 11 years ago, six to samples from 9 years ago, and five to those collected 8 years ago (Figure 1).

Amplification of the human ribonuclease P (RNAseP) gene was successful in all specimens analysed, confirming the presence of amplifiable human DNA in the extracted material. The detection of human DNA after long-term storage confirms both the integrity of the stored sample and the robustness of the extraction protocol in recovering amplifiable DNA under these conditions.

All the studied samples were obtained from female individuals aged between 15 and 37 years. A total of 14 positive results were identified from vaginal samples, six from vulvar samples, and one from a combined perianal and oral sample. With regard to *T. vaginalis*, the pathogen was detected in three vulvar samples, eight vaginal samples, one perianal sample, and one oral sample. In the case of *C. trachomatis*, it was detected in four vulvar samples and seven vaginal samples. One vulvar sample (IST200) tested positive for both *T. vaginalis* and *C. trachomatis*.

In this study, the obtained positive results correspond to 12 individuals with samples collected from different anatomical regions in some cases. The cases were analyzed between 2004 and 2017 as part of the forensic casework of the National Institute of Legal Medicine and Forensic Sciences. Genetic profiles were obtained from each sample, whenever possible, through short tandem repeat (STR) analysis. Regarding autosomal STR, each sample was processed using the GlobalFiler™ PCR Amplification Kit (Thermo Fisher Scientific, Waltham, MA, USA), and for the Y-STR analysis, it was carried out using the Yfiler™ Plus PCR Amplification Kit (Thermo Fisher Scientific, Waltham, MA, USA). It was possible to detect a mixture of female and male genetic profiles in certain individuals, as well as the presence of the Y-chromosome haplotype. In the case of individual 12, as indicated in Table 1, a single male genetic profile was identified.

This study was designed as a descriptive analysis; therefore, no inferential statistical tests were conducted.

## 3. Discussion

Sexual crimes are associated with a wide range of long-term consequences for survivors, among which STIS are particularly significant due to their frequently asymptomatic nature and their potential to cause chronic health complications [4]. Despite their clinical and forensic importance, investigations specifically addressing the detection of STIS in biological evidence from sexual assault cases remain limited. This scarcity of research may contribute to the underestimation of infection rates, diagnostic delays, and missed opportunities for timely medical or forensic intervention.

The retrospective analysis of stored forensic samples for STI microorganism detection also raises important ethical and legal considerations. Reliable identification of pathogens years after the assault may contribute to delayed justice, therapeutic intervention, and epidemiological tracking. Thus, preserving the integrity of biological evidence and applying sensitive molecular techniques are essential for ensuring the rights of survivors and the robustness of forensic conclusions. The biological samples analyzed in the study were collected during forensic medical examinations conducted as part of criminal investigations into sexual assault cases in Portugal. These samples were obtained under judicial authority and used exclusively for forensic purposes, including the identification of perpetrators. According to Portuguese law, biological samples from cases that occurred more than two years ago may be used for research purposes, provided that full anonymity is maintained. All samples included in the study were anonymized prior to analysis, and the corresponding criminal cases had been fully processed and legally closed. The study was approved by the Ethics Committee of the National Institute of Legal Medicine and Forensic Sciences. From a legal standpoint, the ability to detect pathogen DNA with high accuracy even after prolonged storage may introduce new probative elements into previously inconclusive cases, enabling the reconsideration of cold cases and the reopening of judicial proceedings when new scientific evidence corroborates survivor testimony or strengthens forensic links to alleged perpetrators.

The implementation of real-time PCR to detect STI microorganisms in forensic settings represents a significant methodological advancement. This molecular technique was selected for this study due to its multiple advantages, such as high sensitivity and specificity, rapid processing time, reduced risk of cross-contamination, and efficient target amplification [17], and offers the ability to detect pathogen DNA in samples that are often unsuitable for conventional culture-based diagnostics [16]. The real-time PCR approach used in this study enabled the detection of *Chlamydia trachomatis* and *Trichomonas vaginalis* in forensic samples stored for up to 18 years, with prevalence rates of 4.8% and 5.6%, respectively.

These results are consistent with those reported by Sachs et al. (2022), who observed comparable prevalence rates of *C. trachomatis* and *T. vaginalis* among victims of sexual violence [18]. Similar findings have been reported by other studies using PCR-based diagnostics, including those by Kebbi-Beghdadi et al. (2022), Schirm et al. (2007), Caliendo et al. (2005), and Šoba et al. (2015) [19,20,21,22]. In contrast, a notably higher prevalence of *T. vaginalis* was reported by Elsherif et al. (2013), which may be attributable to the preselection of culture-positive samples prior to PCR analysis [23]. These discrepancies in prevalence rates likely reflect variations in sample selection criteria, population characteristics, methodological approaches, and storage or handling conditions [24].

Comparison of our findings with previously published molecular studies highlights considerable heterogeneity in the reported prevalence of *C. trachomatis* and *T. vaginalis*. This heterogeneity underscores the relevance of population-specific and methodological factors in STI detection and reinforces the diagnostic value of molecular tools in forensic casework.

The observed detection rates—particularly in samples stored for up to two decades—emphasize the importance of integrating STI screening into routine forensic protocols. The frequent asymptomatic nature of these infections, combined with extended storage durations between sample collection and analysis, poses a challenge for conventional diagnostic methods. In our study, pathogen DNA was successfully amplified from long-term preserved samples, including IST015 and IST016 from individual 1, collected in 2007, and confirmed positive for *C. trachomatis*. These findings confirm that, when appropriate preservation protocols are followed, real-time PCR remains a reliable tool for STI detection in archived forensic materials.

The results obtained indicate a higher prevalence of *T. vaginalis* and *C. trachomatis* in women aged between 15 and 37 years, which, according to the WHO, is the age group most vulnerable to these infections [1]. A greater number of positive cases was observed in vaginal and vulvar samples, suggesting that these are the primary targets for detecting such infections. Furthermore, the detection of STI microorganisms in perianal and oral samples highlights important considerations regarding the inclusion of these anatomical regions in sampling protocols, particularly in cases of sexual assault.

Simultaneous detection of *T. vaginalis* and *C. trachomatis* was observed in a single vulvar sample (IST200). This finding is of particular significance, as coinfections have the potential to augment the risk of transmission and may complicate therapeutic management for the affected individual. Consequently, these results highlight the importance of implementing multiplex diagnostic approaches, given that the identification of one sexually transmitted infection may be indicative of the concurrent presence of additional pathogens.

According to the study by Silva et al. (2020) [25], the prevalence rates of *N. gonorrhoeae* and *T. pallidum* were 1.3% and 1.0%, respectively. In this context, the absence of positive results for these pathogenic agents in our study may indicate a lower prevalence of these infections in the population investigated. Nonetheless, the potential for DNA degradation over time cannot be excluded, as it may compromise the integrity of the genetic material and, consequently, diminish the sensitivity of molecular diagnostic methods. This consideration emphasizes the critical importance of appropriate preservation and handling of biological specimens, particularly with regard to the stability of nucleic acids, in order to ensure the reliability of diagnostic outcomes—especially in cases where the pathogens are present in low abundance.

It should be noted that, considering the obtained positive results for *T. vaginalis* and *C. trachomatis*, the simultaneous presence of both female and male genetic profiles was observed in some of these samples, including the identification of Y-chromosome haplotypes. This finding confirms the presence of male genetic material, which may be indicative of sexual contact and holds particular relevance in forensic and criminal investigative contexts.

In the specific case of individual 12, an exclusively male genetic profile was identified, which may contribute to the forensic identification of the perpetrator. Similarly, in those individuals in whom a Y-chromosome haplotype was detected, the same evidentiary potential exists, provided that an appropriate reference sample is available for comparative analysis and the identification of a potential suspect. This detection supports the possibility of genetic material transfer between individuals. The presence of multiple genetic profiles—both female and male—also introduces important considerations for the interpretation of results, as the coexistence of DNA from multiple contributors can complicate the individual identification and the forensic attribution of genetic material.

Nonetheless, several limitations must be acknowledged. Although the total number of swabs analysed (*n* = 231) is considerable within the context of forensic casework, the absence of stratified subgroups based on variables such as the age of the victim or the duration of sample storage has limited the scope for comparative analyses. This, in turn, constrains the capacity to identify patterns or associations that may hold relevance for clinical or forensic interpretation.

Additionally, the study did not include inferential statistical analyses, as the retrospective and descriptive nature of the dataset was primarily intended to assess the feasibility of molecular detection of STI microorganisms in long-term stored forensic samples. Future studies should incorporate larger, stratified samples and statistical approaches to assess associations over time or by sample type.

Another important limitation is the absence of direct comparison with conventional diagnostic techniques, such as culture or microscopy. Without this, it is not possible to fully assess the diagnostic accuracy and specificity of real-time PCR in forensic applications. This is particularly relevant for its integration into standard forensic workflows, where complementary diagnostic validation is often necessary.

Nonetheless, previous studies have shown that real-time PCR generally demonstrates higher sensitivity and faster turnaround times compared to conventional culture methods, particularly when applied to samples with low pathogen load or compromised viability [20,26]. These characteristics make PCR particularly advantageous in forensic contexts, where sample integrity may be affected by delayed collection and long-term storage.

Despite these constraints, the present findings support the long-term applicability and forensic value of real-time PCR. This technique has demonstrated robust performance even in archived samples, making it a promising candidate for routine integration into forensic and clinical workflows involving sexual assault cases. Future research should aim to harmonize sample processing protocols, investigate the influence of different storage conditions on diagnostic yield, and validate these results in multicenter and prospective settings.

## 4. Materials and Methods

### 4.1. Sample Collection

A total of 231 biological swabs collected from sexual assault victims between 2004 and 2017 were included in our study. Samples were initially collected only with a focus on aggressor genetic profile detection. So, they were initially studied with that focus and stored at −20 °C in a specific cold storage room for up to 18 years.

Samples were obtained from diverse anatomical locations, including 112 vaginal, 57 vulvar, three peri-vulvar, one perineum, one inner thigh, 32 anal canal, three rectal, 12 perianal, and 10 oral.

These samples were selected taking into account the number of old sexual assault cases available for subsequent analysis in 2024.

### 4.2. DNA Extraction and Purification

The extraction and purification of DNA are critical steps in microbiological investigations, as they directly impact the sensitivity and reliability of downstream molecular analyses [27]. Various protocols—both manual and automated—are available, with selection depending on the type of biological sample, time constraints, and specificity requirements [28].

In this study, DNA was extracted and purified using an automated platform, the QIAGEN EZ1 (QIAGEN, Hilden, Germany), with DSP Virus Kit, according to the manufacturer’s protocol. This kit offers several advantages, including high reproducibility, reduced contamination risk, and efficient nucleic acid recovery [27].

For sample preparation, a portion of each swab containing biological material was cut and transferred into a 1.5 mL microcentrifuge tube containing 1.0 mL of 0.9% sodium chloride (NaCl) solution. Each tube was labeled with the corresponding case identifier. To minimize the risk of cross-contamination, scissors and work surfaces were disinfected with 70% ethanol between the processing of each sample. The tubes were vortexed for approximately 10 s and incubated in a dry heating block at 56 °C for 30 min to inactivate potential pathogens. After inactivation, samples were stored at −20 °C until DNA extraction, which was performed using the BioRobot EZ1 Advanced automated system (QIAGEN, Hilden, Germany), following the standard protocol. In each extraction, a negative control is used to monitor contamination.

### 4.3. Molecular Detection and DNA Amplification

Real-time PCR enables the detection of bacterial DNA through the use of sequence-specific primers, oligonucleotide probes, and fluorescent reporter molecules. These probes hybridize with the target DNA sequence and emit a fluorescence signal, which produces an amplification curve, indicating the presence of the microorganisms’ DNA [29].

In this study, fluorescence signals were detected using FAM and HEX fluorophores, and pathogen-specific primers were employed for the identification of each targeted sexually transmitted infection. The target genes/sequences, primers, and probes used for the detection of microorganisms are presented in Table 2. In this work, the primers and probes were used at a concentration of 400 nM and 100 nM, respectively.

For each reaction, the PCR master mix was prepared as follows: 12.5 μL of NZYSupreme Multiplex qPCR Master Mix, 5.5 μL of RNase-free water, and 2 μL of the corresponding primer/probe mix.

A total of 20 μL of this master mix was combined with 5 μL of the extracted DNA sample in each reaction tube. Amplification was conducted using the CFX96™ Real-Time PCR Detection System (Bio-Rad, Hercules, CA, USA). For *Trichomonas vaginalis* DNA, the real-time PCR amplification program consisted of an initial hold at 50 °C (2 min), denaturation at 95 °C (10 min), followed by 40 cycles at 95 °C (15 s) and at 60 °C (1 min) [20]. The thermal cycling conditions for *Chlamydia trachomatis* DNA and *Neisseria gonorrhoeae* DNA consisted of an initial hold at 95 °C (2 min), followed by 40 cycles at 95 °C (30 s) and 60 °C (30 s) for *C. trachomatis,* and 55 °C (30 s) for *N. gonorrhoeae* [30]. For *Treponema pallidum* DNA, amplification conditions involved 50 cycles, including an initial denaturation at 95 °C (30 s), followed by 55 °C (30 s) and 72 °C (30 s) [31,32].

In each extraction run, a negative control was used to monitor contamination. The positive controls for each microorganism used in these experiments were certified reference-specific DNA solutions for each microorganism, from a commercial source, to guarantee that the amplification process was successful. To confirm the integrity of the samples, the human RNAseP gene was amplified in all analyzed specimens. RNAseP is widely used as an internal control in molecular diagnostics, serving as a marker of sample adequacy and amplification reliability.

Amplification of RNAseP was carried out using primers and probes specific to the human gene. For each reaction, a master mix was prepared containing 12.5 μL of NZYSpeedy One-step RT-qPCR Probe Master Mix (2×), 6 μL of RNase-free water, and 1.5 μL of RNAseP primers/probes. To this mixture, 5 μL of the extracted sample DNA was added, resulting in a final reaction volume of 25 μL.

The amplification was performed using the CFX96™ Real-Time PCR Detection System (Bio-Rad, CA, USA) under thermal cycling conditions as follows: initial denaturation at 95 °C (2 min), followed by 40 cycles at 95 °C (5 s) and 60 °C (30 s) [33]. The successful amplification of the human RNAseP gene confirmed the presence of human DNA and validated the overall reliability of the molecular workflow. The primer and probe sequences are listed in Table 1.

All PCR assays included positive and negative controls, and a cycle threshold (Ct) value below 35 was considered indicative of a positive result.

Both the extraction and amplification processes were carried out in duplicate for each sample. In cases where discrepancies were observed between duplicates, the samples were repeated to ensure confidence in the results.

## 5. Conclusions

Although we do not have information regarding the detection of STIS in these samples at the time of collection, this study proves to be extremely important due to the fact that we obtained positive STI results after 18 years of storage, in some samples. The successful amplification of pathogen DNA from long-term preserved forensic samples highlights the robustness of real-time PCR and underscores its applicability in contexts where sample integrity is frequently compromised. These findings emphasize the value of molecular diagnostics for retrospective forensic investigations, support clinical decision-making in the care and follow-up of survivors, and contribute to epidemiological surveillance by addressing diagnostic gaps in underreported or unresolved cases.

Real-time PCR proved to be a promising tool in both medical and legal responses to sexual violence. Future research should aim to validate these findings across diverse settings, expand the range of detectable pathogens—including viral agents—and standardize storage and handling protocols. Multicenter, prospective studies stratified by variables such as anatomical sampling site and storage duration will be essential to fully establish the role of molecular diagnostics as a reference method in forensic casework involving sexual assault.

## Figures and Tables

**Figure 1 ijms-26-08124-f001:**
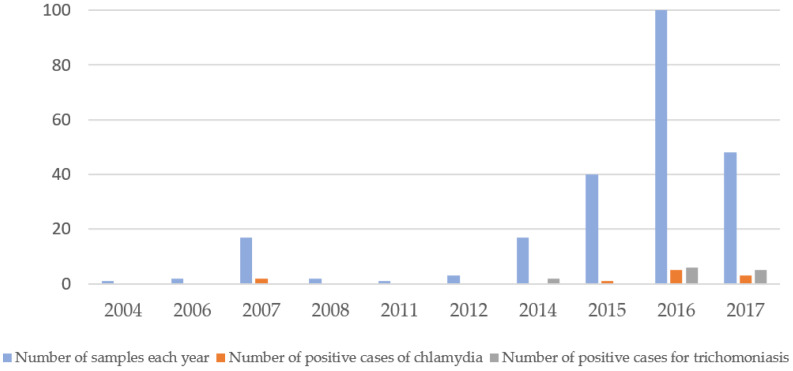
Representation of the number of positive cases for chlamydia (orange bars), the number of positive cases of trichomoniasis (gray bars), and the number of total samples studied per year, between 2004 and 2017.

**Table 1 ijms-26-08124-t001:** Information regarding each alleged sexual assault victim who tested positive for *C. trachomatis* and/or *T. vaginalis*, including year of collection, victim’s age, sample type, STI detected, and genetic profile identification.

Alleged Victim	Year of Collection	Victim’s Age	Sample ID	Swab Type	STI Detected	Genetic Profiles Obtained
Individual 1	2007	22	IST015	Vulvar	*Chlamydia trachomatis*	No information
			IST016	Vaginal	*Chlamydia trachomatis*	
Individual 2	2014	36	IST032	Vulvar	*Trichomonas vaginalis*	No male profile identified
			IST033	Vaginal	*Trichomonas vaginalis*	
Individual 3	2015	37	IST073	Vaginal	*Chlamydia trachomatis*	Female and male admixture
Individual 4	2016	14	IST092	Vulvar	*Trichomonas vaginalis*	No male profile identified
			IST104	Vaginal	*Trichomonas vaginalis*	
Individual 5	2016	Unknown	IST095	Vaginal	*Chlamydia trachomatis*	Female and male admixture
Individual 6	2017	23	IST142	Vaginal	*Chlamydia trachomatis*	Male profile (STR) + 2 different Y haplotype profiles
			IST147	Vulvar	*Chlamydia trachomatis*	
			IST148	Vaginal	*Chlamydia trachomatis*	
Individual 7	2017	26	IST154	Perianal	*Trichomonas vaginalis*	Female and male admixture profile + Y haplotype
			IST158	Vaginal	*Trichomonas vaginalis*	
			IST197	Vaginal	*Trichomonas vaginalis*	
Individual 8	2016	15	IST169	Vulvar	*Chlamydia trachomatis*	Female and male admixture profile + Y haplotype
			IST171	Vaginal	*Chlamydia trachomatis*	
			IST196	Vaginal	*Chlamydia trachomatis*	
			IST200	Vulvar	*Trichomonas vaginalis* and *Chlamydia trachomatis*	
Individual 9	2017	17	IST190	Vaginal	*Trichomonas vaginalis*	No male profile identified
Individual 10	2016	25	IST198	Oral	*Trichomonas vaginalis*	Admixture profile of two Y haplotypes in a sample
Individual 11	2017	31	IST199	Vaginal	*Trichomonas vaginalis*	Male profile (STR) + Y haplotype
Individual 12	2016	37	IST204	Vaginal	*Trichomonas vaginalis*	Male profile identified
			IST232		*Trichomonas vaginalis*	

**Table 2 ijms-26-08124-t002:** Target genes/sequences, primers, and probes for the detection and amplification of microorganisms and human RNAseP [20,30,31,32,33].

Target Gene	Primer Sequences (5′3′)
Cryptic plasmid from *C. trachomatis*	F: AACCAAGGTCGATGTGATAG R: TCAGATAATTGGCGATTCTT P: ROX-CGAACTCATCGGCGATAAGG
Por A pseudogene from *N. gonorrhoeae*	F: CCGGAACTGGTTTCATCTGATT R: GTTTCAGCGGCAGCATTCA P: CGTGAAAGTAGCAGGCGTATAGGCGGACTT
polA gene from *T. pallidum*	F: GGTAGAAGGGAGGGCTAGTA R: CTAAGATCTCTATTTTCTATAGGTATGG P: ACACAGCACTCGTCTTCAACTCC
2 kb repeat sequence from *Trichomonas vaginalis*	F: AAG ATG GGT GTT TTA AGC TAG ATA AGG T R: CGT CTT CAA GTA TGC CCC AGT AC P: CCG AAG TTC ATG TCC TCT CCA AGC GT
Human RNase P gene	F: AGATTTGGACCTGCGAGCG R: GAGCGGCTGTCTCCACAAGT P: TTCTGACCTGAAGGCTCTGCGCG

## Data Availability

The original contributions presented in this study are included in the article. Further inquiries can be directed to the corresponding author.

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
