# Peer review of "Sexually Transmitted Infections: Usefulness of Molecular Methods for Microorganism Detection in Stored Sexual Assault Samples"

_ijms, 2025, doi:10.3390/ijms26178124_

Round 1
Reviewer 1 Report (Previous Reviewer 1)
Comments and Suggestions for Authors
As with the previously submitted manuscript, Caine et al utilize real-time PCR to detect four sexually transmitted pathogens in samples taken from victims of sexual assault and stored for up to 18 years. By analyzing a highly relevant collection of historic samples using a gold standard molecular detection method, the study provides evidence that such samples can still yield potentially significant forensic findings many years after collection.
The new manuscript is substantially improved from the previous submission, and many of the points raised previously have been dealt with appropriately. My only comment would be regarding Table 3. As the text makes clear (line 247), only sample IST200 tested positive for both pathogens, yet Table 3 suggests that all the samples from individual 8 (ISTs169, 171, 196 and 200) tested positive for both organisms. In Table 3, the STI result of each sample ID should be provided, not the overall result per individual, and any discrepancies (if there are any) - for example, if only one out of two vaginal swabs from the same individual tested positive for an STI), briefly discussed.
Author Response
Comments: As with the previously submitted manuscript, Caine et al utilize real-time PCR to detect four sexually transmitted pathogens in samples taken from victims of sexual assault and stored for up to 18 years. By analyzing a highly relevant collection of historic samples using a gold standard molecular detection method, the study provides evidence that such samples can still yield potentially significant forensic findings many years after collection. The new manuscript is substantially improved from the previous submission, and many of the points raised previously have been dealt with appropriately. My only comment would be regarding Table 3. As the text makes clear (line 247), only sample IST200 tested positive for both pathogens, yet Table 3 suggests that all the samples from individual 8 (ISTs169, 171, 196 and 200) tested positive for both organisms. In Table 3, the STI result of each sample ID should be provided, not the overall result per individual, and any discrepancies (if there are any) - for example, if only one out of two vaginal swabs from the same individual tested positive for an STI), briefly discussed.
Answer: We thank the reviewer for this valuable suggestion. The suggested changes were applied in Table 3
Reviewer 2 Report (New Reviewer)
Comments and Suggestions for Authors
The study from Caine et al. presents a well-executed retrospective analysis of archived biological samples from sexual assault cases, demonstrating the long-term applicability of real-time PCR for detecting Chlamydia trachomatis and Trichomonas vaginalis especially. The work is highly relevant to both forensic and clinical fields and contributes meaningfully to the existing literature.
The following suggestions for revision -
1. Clarify Pathogen-Specific Sensitivity. The absence of positive results for Neisseria gonorrhoeae and Treponema pallidum is interesting, and needs further discussion. First, in Table 1 (as I'm no expert), what are the pathogen-specific targets of these primers? Second, what are the positive controls for the pathogen-specific experiments - these need to be clarified (are they samples of laboratory strains?) Did the authors confirm the assay performance with these controls or ideally with spiked swab samples? Assay performance is critical, especially in this forensic setting. Evidence of assay performance of the controls would be good to have in supplementary.
2. In the context of no gonococci or syphilis, could degradation or lower initial pathogen load explain the results, or is there population-specific low prevalence (I am not up to date with STIs in Portugal, but one might expect lower prevalence of syphilis, but not gonorrhoea? -Might be useful to add a note on cases of these pathogens over the years of the sample study, for context).
3. Storage Conditions Detail - this is obviously important and a limitation of the study is the fact that there is not much information as to the storage of these samples. Do the authors know is they were samples continuously stored at −20 °C without thawing, and was there any temperature monitoring and recording? This might be important for forensic purposes.
4. Ethical and Legal Implications - there are obviously ethical/legal implications around the retrospective testing of samples. perhaps some discussion of these could be added to the discussion.
5. Data presentation: I would suggest that the data in Fig 1 and Fig 2 are combined into a Table with the yearly number of cases broken down. It is easier to digest the relative numbers than following the text of trying to read the Figures.
Also, the supplementary Tables of PCR conditions are unnecessary; these usually go in methods as running text (please place them there).
6. General comment - again, I'm no expert in forensics, but in the case of rape, sexual offences, are fresh swab samples not taken and analysed for STIs and included in 'rape kits'? Might be worth a comment.
7. Minor revisions -some typographical errors, e.g. line 282, offers not offres.
8. References -update WHO reference, Farahi reference to 2025.
N/A
Author Response
1.Clarify Pathogen-Specific Sensitivity. The absence of positive results for Neisseria gonorrhoeae and Treponema pallidum is interesting, and needs further discussion. First, in Table 1 (as I'm no expert), what are the pathogen-specific targets of these primers? Second, what are the positive controls for the pathogen-specific experiments - these need to be clarified (are they samples of laboratory strains?) Did the authors confirm the assay performance with these controls or ideally with spiked swab samples? Assay performance is critical, especially in this forensic setting. Evidence of assay performance of the controls would be good to have in supplementary.
Answer: We thank for the reviewer comment. We modified the table 1 including the pathogen-specific targets. We modified the line 319 for inclusion of the pathogen-specific targets information.
(Line 313-314) The target genes/sequences, primer and probes used for the detection of microorganisms are presented in Table 2.
In addition, we clarified now in the manuscript the origin of our positive controls.
(Line 332-334) The positive controls for each microorganism used in these experiments were certified reference specific DNA solutions for each microorganism, from a commercial source, and allow to guarantee that the amplification process was successful.
- In the context of no gonococci or syphilis, could degradation or lower initial pathogen load explain the results, or is there population-specific low prevalence (I am not up to date with STIs in Portugal, but one might expect lower prevalence of syphilis, but not gonorrhoea? -Might be useful to add a note on cases of these pathogens over the years of the sample study, for context).
Answer: We really appreciate your suggestion. We agree with the reviewers that there may be several causes for the results obtained for these pathogens, namely the degradation of the sample's genetic material as well as the fact that these pathogens have a lower prevalence in the population. The information about the prevalence of gonorrhea and syphilis has now been added to the manuscript.
(Line 222-225) According to the study by Silva et al. (2020), the prevalence rates of N. gonorrhoeae and T. pallidum were 1.3% and 1.0%, respectively. In this context, the absence of positive results for these pathogenic agents in our study may indicate a lower prevalence of these infections in the population investigated.
- Storage Conditions Detail - this is obviously important and a limitation of the study is the fact that there is not much information as to the storage of these samples. Do the authors know is they were samples continuously stored at −20 °C without thawing, and was there any temperature monitoring and recording? This might be important for forensic purposes.
Answer: We thank for the reviewer comment. The sample set of 231 samples were continuously stored at −20 °C until the processing in specific cold storage room of the National Institute of Legal Medicine and Forensic Sciences of Portugal.
(Line 278-280) Samples were initially collected only with focus on aggressor genetic profile detection. So, they were initially studied with that focus and stored at -20ºC in specific cold storage room for up to 18 years.
- Ethical and Legal Implications - there are obviously ethical/legal implications around the retrospective testing of samples. perhaps some discussion of these could be added to the discussion.
Answer: We thank for the reviewer comment. We add some sentences addressing this issue to the discussion section as follows:
(Line 162-175) The biological samples analyzed in the study were collected during forensic medical examinations conducted as part of criminal investigations into sexual assault cases in Portugal. These samples were obtained under judicial authority and used exclusively for forensic purposes, including the identification of perpetrators. According to Portuguese law, biological samples from cases that occurred more than two years ago may be used for re-search purposes, provided that full anonymity is maintained. All samples included in the study were anonymized prior to analysis, and the corresponding criminal cases had been fully processed and legally closed. The study was approved by the Ethics Committee of the National Institute of Legal Medicine and Forensic Sciences.
From a legal standpoint, the ability to detect pathogen DNA with high accuracy even after prolonged storage may introduce new probative elements into previously inconclusive cases, enabling the reconsideration of cold cases and the reopening of judicial proceedings when new scientific evidence corroborates survivor testimony or strengthens forensic links to alleged perpetrators.
- Data presentation: I would suggest that the data in Fig 1 and Fig 2 are combined into a Table with the yearly number of cases broken down. It is easier to digest the relative numbers than following the text of trying to read the Figures.
Also, the supplementary Tables of PCR conditions are unnecessary; these usually go in methods as running text (please place them there).
Answer: We thank for the reviewer comment. We think the data looks more appealing in a figure, so figures 1 and 2 have been combined into one, representing figure 1, thus following one of the suggestions given by reviewer 3.
As suggested we remove the supplementary tables of PCR and included this information at the manuscript in material and methods section as follows:
(Line 324-331) For Trichomonas Vaginalis DNA, the real-time PCR amplification program consisted in an initial hold at 50ºC (2 min), denaturation at 95 ºC (10 min), followed by 40 cycles at 95 ºC (15 s) and at 60ºC (1 min) [20]. The thermal cycling conditions for Chlamydia trachomatis DNA and Neisseria gonorrhoeae DNA consisted in an initial hold at 95ºC (2 min), followed by 40 cycles at 95 ºC (30 s) and 60ºC (30 s) for C. trachomatis, and 55ºC (30 s) for N. gonorrhoeae[30]. For Treponema Pallidum DNA, amplification conditions involved 50 cycles included an initial denaturation at 95 ºC (30 s), followed by 55 ºC (30 s) and 72ºC (30 s) [31; 32].
(Line 345-346) …as follows, initial denaturation at 95ºC (2 min), followed by 40 cycles at 95 ºC (5 s) and 60ºC (30 s) [33].
- General comment - again, I'm no expert in forensics, but in the case of rape, sexual offences, are fresh swab samples not taken and analysed for STIs and included in 'rape kits'? Might be worth a comment.
Answer: We appreciate the reviewer 's relevant comment. The samples included in the present study were collected during forensic medical examinations conducted as part of criminal investigations into sexual assault cases and used only for forensic purposes. The samples were not tested for STI at the time of collection. According to Portuguese law, biological samples from cases that occurred more than two years ago, such as the samples included in our study, may be used for research purposes.
We have included new information regarding this issue to the discussion section. Please see our response to the reviewer's comment 4.
- Minor revisions -some typographical errors, e.g. line 282, offers not offres.
Answer: We greatly appreciate the suggestion. This change was made (line 180)
- References -update WHO reference, Farahi reference to 2025.
Answer: Thank you very much for the suggestion. The change was made.
Reviewer 3 Report (New Reviewer)
Comments and Suggestions for Authors
This interesting report for a special issue in this Journal concerns the detection of several sexual transmitted infection (STI) pathogens in (mostly) stored samples (N=231), collected during investigations in cases of sexual assault. The time period of collection was several years, from 2004 to 2017. Mainly Chlamydia trachomatis (CT) and Trichomonas vaginalis (TV) sequences were found using PCR.
The manuscript itself was unsual because is contained tracked changes, making it less convenient to read. Why was not a clean copy of the manuscript submitted?
Another general remark is that many text passages are general education remarks. These do not belong in a study report. The total manuscript can easily be reduced to a short paper.
There are major and minor points that should be addressed to further improve this manuscript.
Major comments.
- A lot of general theory is given. Since this is not a book chapter but an article for a scientific journal, describing a certain analysis, I suggest to delete these general, irrelevant paragraphs. Specifically the following paragraphs and lines can be deleted in
Introduction: lines 46-54; lines 81-87; lines 107- 122
and Methods: lines195-200. Other paragraphs may also be shortened.
Please adjust.
- Tables 1 and 2 can be combined, so the primers and probe used for the human RNaseP gene can be presented in the same table as those for detection of the STI pathogens. Please also mention all target genes and regions for each of the primer/probe sets.
- There are far too many Figures. The pie-chart in Figure 1 is not needed since it just says the same as is mentioned in the text: 13 positives for TV and 11 for CT. Actually all results can be shown in one figure instead, in which you show per calendar year three bars: total number of samples, number of CT positives and number of TV positives. If needed the positives can be shown as %, with an extra Y-axis. Please also remove the redundant text like how many years ago collected (lines 269 -273). This will be obvious from the Figure.
- Table 3 and the text describing this table (lines 291- 300) is interesting. However a number of acronyms are now introduced in Results that were not explained earlier. Examples are: SGBF, INMLCF, STR analysis, Y-STR. These should all be given in full and be described in Methods how typing was performed. Please adjust.
- Table 4 in the Discussion does not contain any studies on sexual assault thus these references are not very relevant for this study. In addition, ‘clinical samples’ and ‘culture-positive pre-selected samples’ are not ‘study populations’. Please delete this Table.
- The Discussion is very (too) long. Please just discuss relevant issues pertaining to the detection of the bacterial pathogens in sexual assault samples. Avoid redundancies (for example lines 420-426).
- The conclusions is now a full paragraph. This should be one or two sentences at most. Please adjust.
Minor, editorial comments.
- Abstract, line 27: please replace ‘in’ by ‘for’ before ‘some samples’
- Abstract, line 32: I suggest to replace ‘sensitivity and reliability’ by ‘usefulness’, because in this analysis no comparisons with other diagnostic methods were made. PCR was simply performed on the collected samples and if positive, this was a demonstration that PCR can be useful. Please adjust.
- Material and Methods, line 164: please delete the referral to Figure 1. Figure 1 is part of Results and is not paced in Methods.
- Lines 187/188: I suggest to write ‘in each extraction run a negative control was used to monitor contamination’.
- With the ‘track changes’ version it is hard to read but it looks like in the heading of Table 1 references [23-26] are given. This is unusual. It would be better to mention the references in the legend of the Table. Please adjust.
Most of the text is well written but there are parts that should be improved. I suggest to drastically shorten the text. Please take care that the new texts are written in good Eglish grammar and spelling.
Author Response
Major comments.
1.A lot of general theory is given. Since this is not a book chapter but an article for a scientific journal, describing a certain analysis, I suggest to delete these general, irrelevant paragraphs. Specifically the following paragraphs and lines can be deleted in
Introduction: lines 46-54; lines 81-87; lines 107- 122 and Methods: lines195-200. Other paragraphs may also be shortened.
Answer: Thank you very much for the suggestion. Line 46-54; 81-87 and 107- 122 has been removed
2.Tables 1 and 2 can be combined, so the primers and probe used for the human RNaseP gene can be presented in the same table as those for detection of the STI pathogens. Please also mention all target genes and regions for each of the primer/probe sets.
Answer: We greatly appreciate the suggestion. The tables were combined into a single one, resulting in Table 1. The genes and target regions for each primer set have also been added to this table.
3.There are far too many Figures. The pie-chart in Figure 1 is not needed since it just says the same as is mentioned in the text: 13 positives for TV and 11 for CT. Actually all results can be shown in one figure instead, in which you show per calendar year three bars: total number of samples, number of CT positives and number of TV positives. If needed the positives can be shown as %, with an extra Y-axis. Please also remove the redundant text like how many years ago collected (lines 269 -273). This will be obvious from the Figure.
Answer: Thank you very much for your suggestion. The figures have been combined into one including the total number of samples, number of CT positives and number of TV positives in three bars, resulting in Figure 1. The redundant text has also been removed.
4.Table 3 and the text describing this table (lines 291- 300) is interesting. However a number of acronyms are now introduced in Results that were not explained earlier. Examples are: SGBF, INMLCF, STR analysis, Y-STR. These should all be given in full and be described in Methods how typing was performed. Please adjust.
Answer: Thank you very much for your suggestion. It was a mistake. The full names of each acronyms are now included in the manuscript.
5.Table 4 in the Discussion does not contain any studies on sexual assault thus these references are not very relevant for this study. In addition, ‘clinical samples’ and ‘culture-positive pre-selected samples’ are not ‘study populations’. Please delete this Table.
Answer: Thank you very much for the suggestion. Table 4 has been deleted.
6.The Discussion is very (too) long. Please just discuss relevant issues pertaining to the detection of the bacterial pathogens in sexual assault samples. Avoid redundancies (for example lines 420-426).
Answer: Thank you very much for your comment. We tried to reduce the discussion section following your suggestions. However, another reviewer asked us to reinforce some other relevant aspects of this section, so we made some changes.
7.The conclusions is now a full paragraph. This should be one or two sentences at most. Please adjust.
Answer: We greatly appreciate the suggestion. The conclusion has been adjusted and reduced, as follows:
(line 356-370) Although we do not have information regarding the detection of STI in these samples at the time of collection, this study proves to be extremely important due to the fact that we obtained positive STI results after 18 years of storage, in some samples. The successful amplification of pathogen DNA from long-term preserved forensic samples highlights the robustness of real-time PCR and underscores its applicability in contexts where sample integrity is frequently compromised. These findings emphasize the value of molecular diagnostics for retrospective forensic investigations, support clinical decision-making in the care and follow-up of survivors, and contribute to epidemiological surveillance by addressing diagnostic gaps in underreported or unresolved cases.
Real-time PCR proved to be a promising tool in both medical and legal responses to sexual violence. Future research should aim to validate these findings across diverse settings, expand the range of detectable pathogens—including viral agents—and standardize storage and handling protocols. Multicenter, prospective studies stratified by variables such as anatomical sampling site and storage duration will be essential to fully establish the role of molecular diagnostics as a reference method in forensic casework involving sexual assault.
Minor, editorial comments.
1.Abstract, line 27: please replace ‘in’ by ‘for’ before ‘some samples’
Answer: Thank you very much for the suggestion. The replacement was made.
2.Abstract, line 32: I suggest to replace ‘sensitivity and reliability’ by ‘usefulness’, because in this analysis no comparisons with other diagnostic methods were made. PCR was simply performed on the collected samples and if positive, this was a demonstration that PCR can be useful. Please adjust.
Answer: We greatly appreciate the suggestion. The replacement was made.
(line 32) … demonstrate the usefulness of real-time PCR for detecting STI in long-term preserved forensic samples.
3.Material and Methods, line 164: please delete the referral to Figure 1. Figure 1 is part of Results and is not paced in Methods.
Answer: Thank you for your valuable comment. Reference to figure 1 has been removed
4.Lines 187/188: I suggest to write ‘in each extraction run a negative control was used to monitor contamination’.
Answer: Thank you for your valuable comment. We made the alteration as suggested.
(Line 332-335) In each extraction run a negative control was used to monitor contamination. The positive controls for each microorganism used in these experiments were obtained from laboratory strain samples, and allow to guarantee that the amplification process was successful.
5.With the ‘track changes’ version it is hard to read but it looks like in the heading of Table 1 references [23-26] are given. This is unusual. It would be better to mention the references in the legend of the Table. Please adjust.
Answer: Thank you for your valuable comment. The suggestions were applied.
(line 316-317) Table 2- Target genes/sequences, primer and probes for the detection and amplification of microorganisms and human RNAseP [20; 30; 31; 32; 33].
Round 2
Reviewer 3 Report (New Reviewer)
Comments and Suggestions for Authors
The manuscript has substantially improved. The only comment left is that the Methods are now presented after the Results. Is this according to the wish of the Journal?
Author Response
Dear Reviewer,
Thank you very much for your thoughtful comment.
According to the journal’s guidelines, the recommended structure for manuscripts is as follows:
Introduction
Results
Discussion
Materials and Methods
Conclusions
Therefore, the current order of sections in the manuscript is aligned with the journal’s formatting requirements.
This manuscript is a resubmission of an earlier submission. The following is a list of the peer review reports and author responses from that submission.
Round 1
Reviewer 1 Report
Comments and Suggestions for Authors
The manuscript by Caine et al utilizes real-time PCR to detect four sexually transmitted pathogens in samples taken from victims of sexual assault and stored for up to 18 years. By analyzing a highly relevant collection of historic samples using a gold standard molecular detection method, the study provides evidence that such samples can still yield potentially significant forensic findings many years after collection (if stored under the correct conditions). The study is novel and sound, the manuscript needs significant work to improve clarity, conciseness and readability. Specifically:
- The introduction is overly long and contains non-essential information which could easily be deleted (g. lines 56-62) or text that would be better placed in the discussion (e.g. line 135-139).
- Lines 126-127. Reference the “few studies [which] have systematically evaluated the long-term viability of stored biological samples for molecular testing in sexual assault cases”.
- Section 2.1. Please specifically describe the conditions under which the samples have been stored since collection, as any differences between samples could have a major impact on the study findings.
- Line 155 – specify how many victims the 231 samples came from.
- Lines 159-162. Can be deleted.
- Lines 183-192. Can be moved to discussion.
- Line 197-208. The primers/probe sequences would be better presented in a Table. Also, how were the primers designed? If they have been previously reported/utilized by other researchers, please provide the references.
- Line 211. What concentration were the primers/probe used at?
- Section 2.4. This can be integrated into section 2.3 since it is also information regarding detection and amplification.
- Line 234 and Figure 2. Figure 2 implies that more swabs were available from 2014-2017 than earlier years for testing. Were all the swabs that were available analyzed or were only a sub-set analyzed? If the latter, how were they selected? Why were only samples up to and including 2017 tested?
- Lines 235-237. Please provide a more detailed breakdown of the number of samples from each anatomical site.
- Figure 1 is unnecessary and adds nothing to the statements made in the text.
- Line 262-3. “one perianal/oral sample” – this is not clear, do you mean, one perianal sample and one oral sample?
- Line 267-269. Male/female genetic profiles. No information is provided in the manuscript on how (and when) these profiles/haplotypes were determined.
- Table 1 legend – This needs to clarify that the information shown is only regarding individuals that tested positive for C. trachomatis and/or Trichomonas.
- Table 1 could be improved – it implies that all samples from an individual (e.g. IST154, 158 and 197 from Individual 7) were all positive for a given pathogen (in that example, T. vaginalis). The result of each sample ID should be provided, not the result per individual.
- Again, Table 1 – what is the rationale of including the age of the victim? More relevant information to include would be the year.
- Line 314 – “two cases” – do you actually mean two samples from the same individual?
- Table 2 – positive cases column can be removed. It would be more useful to include the country in which the study was conducted as presumably they were not all in Portugal.
- Figure 4 – if included, this is a result and should be placed in the results section not the discussion, however it does not add much to the manuscript.
- Lines 335-342. Do the authors have epidemiological information on the prevalence of gonorrhoea and syphilis in Portugal compared to the other pathogens between 2004-2017, i.e. were the numbers of cases of these infections generally much lower in the sexually active population than C. trachomatis and Trichomonas meaning that it would be unlikely to detect any positives in just 231 samples anyway?
- Line 343 – “twelve individuals analysed” – do you mean “twelve individuals who tested positive for either C. trachomatis and/or Trichomonas"?
- Lines 388-391. This should be softened since the authors have no way of knowing how many samples would have tested positive at the time of collection.
- Section 5. This should be condensed to 1 or 2 paragraphs.
- Supplementary Tables 1-5. Please add references for the amplification conditions, if previously published.
Reviewer 2 Report
Comments and Suggestions for Authors
In this study, the authors utilize qPCR for detecting STI-causing bacteria such as Chlamydia sp., Trichomonas, Neisseria, and Treponema sp. from 12 individuals and 231 swabs that were stored for 18 years. The qPCR results revealed 13 positive cases of T. vaginalis and 11 for C. trachomatis. The authors claim that real-time qPCR can be utilized extensively for detecting STIs and can help in identifying sexual crimes effectively.
As a summary, the paper seems very incomplete, and the results shown are very limited and need extensive work before it can be accepted for publication. I would reject this paper in the current version for publication. I would strongly encourage the authors to add more results and refine the paper if it has to be considered for publication.
Comments:
- The number of individuals used is not mentioned until we reach Table 1. It is important to mention the sample size (n=12) in the Materials and Methods section.
- Even though in the paper the authors mentioned n=231, it has been taken only from 12 individuals. This means n=12. It is important to clearly mention and differentiate this from each other.
- Among the 12 individuals, how many of the 231 swabs belong to each individual? A clear cutoff of the swabs collected per individual and the number of samples used for qPCR should be provided.
- The authors did not give credit or mention the source from where the victim swabs were collected in lines 155 to 158 under Sample Collection. It is very important to mention this.
- Lines 24 to 33 have been repeated twice in the paper. For example, the same content can be seen in lines 234 to 241. Either remove one of them or rewrite to avoid redundancy.
- Utilization of qPCR for STI diagnosis is already well known, and companies such as Roche are already doing this. What is the novelty of promoting qPCR in this work? In my opinion, testing the integrity of the DNA in the sample stored for 18 years is the only new thing presented here and not checking the sensitivity or specificity using qPCR itself.
- References are missing in lines 73 to 79. More references must be included regarding the use of qPCR for STI diagnostics in general. In another example in lines 114 to 117, the authors state that there are several studies but cite only one reference in line 117. Even if it is a review paper, it is important to provide proper citations of the actual work that indicate material dryness and low DNA recovery rather than just citing a review paper. References also need to be included for lines 141 to 151.
- Usually, the recommended time to identify sexual crimes and STIs is within 72 hours. What is the point of checking it after 18 years? Sure, it could be helpful to check for DNA integrity and the effectiveness of sample storage, but practically there is no motivation as to why a sample needs testing for STI determination or forensic analysis after 18 years. If there is a specific motivation, please discuss.
- There is no discussion about the interference of human DNA with bacterial DNA amplification. The authors discuss amplifying human DNA and bacterial DNA, but what strategies were taken to prevent human DNA from giving background or non-specific amplification? Where are the results? Please include them.
- The authors have mentioned the primers for each of the microorganisms from lines 197 to 208, but how did they get these sequences? How were they validated? Were they taken from a paper? Then it has to be cited. If they were designed using BLAST tools or Primer3, share more information.
- What is the initial concentration of bacterial and human DNA extracted from these swabs? Was it checked using a Nanodrop, including the purity of the sample? Include all this information in the manuscript.
- In line 221, the authors mention that primers and probes specific to a human gene were used. Which human gene? What are the primer and probe sequences? Add this information to the manuscript including the qPCR results.
- The paper mentions 13 positive results for vaginalis and 11 for C. trachomatis. These results are for which individual? Since there are only 12 individuals, number 13 most likely refers to swab number. When 13 and 11 were tested positive, which regions were the swabs taken from? Did this coincide with the actual test and report provided after 72 hours from the victim case? There is no validation of the qPCR results by the authors with actual published reports from each of the victim cases.
- The authors never mentioned what the positive and negative samples looked like for the qPCR assay. Were the primer sequences used for qPCR validated using PCR amplification with positive samples for each organism? If yes, where is this result?
- What is the use of Fig. 1? It does not provide any significant information. It is better to remove this figure.
- For Fig. 2, how many qPCR amplifications were performed per individual per swab? Ideally, three amplification tubes per individual per swab need to be tested to account for reproducibility. Where are these results?
- In lines 247 to 251, the authors suddenly talk about samples tested 10 years ago, five samples 9 years ago, and so on. What was the logic behind choosing these years, and why are there so many random sample sizes chosen for each year? This looks very vague, and no explanation is given.
- In line 255, the authors say it confirms the integrity of the extraction process. How is that?. The only thing it shows is the presence of human DNA after storage for a long time. This shows the integrity of sample storage and DNA in the sample but not of the extraction process. This can be done using any available DNA extraction kit.
- What is the purpose of Fig. 3 if there is no confirmation from other techniques like bacterial plating or published results from the victims?
- In Table 1, all samples were tested after 18 years. If so, why do the authors mention in line 314 that two cases were collected in 2007? From these two individuals mentioned in line 314, how many swabs were tested? Very confusing and abruptly presented results.
- In Table 2, the authors try to compare with existing studies, but after how many years of storage were those samples tested? Add that information for each study.
- Figure 4 is unfortunately unscientific and provides no useful information. The authors, after performing qPCR in 12 individuals with 231 swabs stored for 18 years and including some collected 19 and 9 years prior, show a random graph with a bunch of lines in Fig. 4 without any legends for one individual. It does not show any significance of the test or result. All the qPCR data from 12 individuals from 231 swabs with n=3 for each of these cases (for reproducibility check) should be presented.
- As the authors clearly mention themselves in the discussion section regarding the limitations of this study, without proper validation with plating or the published results obtained when the assault happened, none of these results can be validated.